# Do government innovation preferences enhance ecological resilience in resource-based cities?—Based on mediating effect and threshold effect perspectives

**Jie Zhang, Jiahui Yang** *, Feng Zhao

College of Economics and Management, Shandong University of Science and Technology, Qingdao, China

* 15136858260@163.com

**Data Availability Statement:** All relevant data are within the manuscript and its Supporting information files.

## Abstract

The ecological resilience of resource-based cities reflects the resilience of the ecological environment in resource-based areas to recover after resource development activities, and good ecological resilience holds great significance for the sustainability of the development of resource-based cities. Government innovation preferences are a solid foundation for implementing an innovation-driven strategy and an important guarantee for realizing an efficient production mode. (the purpose of the study) Therefore, to investigate whether government innovation preferences can promote the level of ecological resilience in resource-based cities. (sample information and empirical methods) This paper establishes a mediating effect model and a threshold effect model and empirically analyzes 2009–2020 panel data covering 113 resource-based cities in China as an example. (main findings) The results show the following: (1) the primary term of government innovation preferences has a positive effect on the ecological resilience of resource-based cities, and the secondary term has a negative effect, with a decreasing marginal effect. Additionally, the educational level has a mediating effect on the mechanism of the impact of government innovation preferences on the ecological resilience of resource-based cities. (2) The role of government innovation preferences in the ecological resilience of resource-based cities is heterogeneous: The impact of government innovation preferences on the ecological resilience of resource-based cities in the western region is stronger than that in the central region, and the impact of government innovation preferences on the ecological resilience of medium-sized resource-based cities is stronger than that of large resource-based cities. (3) The role of environmental decentralization produces a single threshold effect with a threshold value of 2.3993 in the impact of the mechanism of government innovation preferences on the ecological resilience of resource-based cities. (contributions and policy implications) The results of this paper can provide theoretical support for the government to set a reasonable innovation preference intensity, and they provide a practical guarantee for the central government to give more environmental governance power to local governments.

**Funding:** This work was supported by the National Social Science Fund of China (No.22BTJ071). The funders had no role in study design, data collection and analysis, decision to publish, or preparation of the manuscript.

**Competing interests:** The authors have declared that no competing interests exist.

# 1. Introduction

The rough industrial development model with high energy consumption, high emissions and high pollution has dominated China's industrial structure for a long period to realize the rapid development of the national economy. However, the continuous seizure and uncontrolled consumption of resources have simultaneously led to serious ecological damage, and ecological resilience has become an important criterion for measuring a region's ecological environmental protection ability. Under the background of strengthening ecological civilization construction in China, there is an urgent need to improve the working mechanism of ecological environmental protection and ecological resilience, especially for resource-based cities, with a deeply rooted dependence on resource exploitation. Changing development thinking, making use of national support for enterprises to implement the policy of green technological innovation, improving the efficiency of resource utilization, and rationally carrying out resource exploitation have become particularly important. Government innovation preferences are an important guarantee for the realization of China's innovation-driven development strategy, and the government's investment in scientific and educational activities can effectively improve the efficiency of ecological environmental protection, thus better helping complete the relevant initiatives for ecological civilization construction in China. In addition, in the context of China's ecological civilization construction, the "centralization" and "decentralization" of power between local governments and the central government have always been a topic of debate, as a certain degree of autonomy for local governments is conducive to more efficient ecological environmental protection. However, an excessive distribution of power may lead local governments to improve the status quo of economic development at the expense of the environment. Therefore, it is important to rationalize the distribution of local governments' power over the environment. Therefore, the research objective of this paper is to establish an indicator measurement system for government innovation preferences, the ecological resilience of resource-based cities, and environmental decentralization by collecting relevant data on 113 resource cities in China from 2009 to 2020. Additionally, relevant hypotheses are proposed, and a mediating effect model and threshold effect model are established to explore the mechanism of government innovation preferences on the ecological resilience of resource-based cities to further provide a theoretical basis and practical guidance for the sustainable development of resource-based cities.

The research motivation, research questions, and research contributions of this paper are as follows. First, the motivation of this paper is the fact that China needs to realize the grand goal of high-quality development, and as a core element of high-quality development, green development is crucial for balancing economic development and ecological protection. However, the model of resource-based cities, in which they seize natural resources to enhance economic development, hurts ecological environmental protection, and the government must improve the ecological resilience of these cities by upgrading the level of production technology and environmental governance innovation in them and by enhancing the degree of freedom of local environmental governance. Therefore, this paper conducts a study on the impact of government innovation preferences on the ecological resilience of resource-based cities under the threshold of environmental decentralization. Second, this study mainly addresses the improvement path of ecological resilience in resource-based cities. Through the government's enhanced investment in innovation, resource-based cities can enhance their level of ecological resilience by upgrading the educational level to cultivate high-tech talent, further optimizing the production mode of high-polluting enterprises and enhancing the efficiency of government environmental governance. Finally, by studying the impact of government innovation preferences on the ecological resilience of resource-based cities under the threshold of

environmental decentralization, this study provides not only a visual representation of the current development of the level of ecological resilience of China's resource-based cities based on the data but also new ideas for government departments to formulate initiatives to improve the level of ecological resilience of resource-based cities based on the conclusions of this study.

The theoretical significance and practical significance of this paper are as follows: The theoretical significance of this paper mainly includes the following: (1) This paper constructs an evaluation mechanism for the ecological resilience of resource-based cities. This study combines the development characteristics of China's resource-based cities and establishes an index evaluation system for the ecological resilience of such cities based on three subsystems: ecosystem resistance, adaptive capacity, and restorative capacity. (2) This paper enriches theory in resource-based cities in regard to realizing the coordinated development of the economy and environmental protection. The government's innovation expenditure has an impact on economic development, and ecological resilience can reflect the strength of environmental protection to a certain extent. This study also introduces the threshold effect perspective of environmental decentralization, which can further fill the gap in research on the coordinated development of economic development and environmental protection in resource-based cities. The practical significance of this paper mainly includes the following: (1) This study provides a new direction for the government to improve the ecological resilience of resource-based cities. (2) The government should apply innovation preference inputs to improve the educational level to enhance the accumulation of innovative talent to contribute to improving ecological resilience. (3) The government should coordinate the relationship between "centralization" and "decentralization", and this study also provides a practical guarantee for the central government to give local governments more power in environmental governance.

The structure of this paper is as follows: The first section summarizes and analyzes the current status of academic research in terms of the relevant theories of this paper. The second section presents the research hypotheses and model construction, proposing hypotheses related to the mechanism of the impact of government innovation preferences on the green development of resource-based cities and establishing a mediating effect model with the educational level as the mediator variable and a threshold effect model with environmental decentralization as the threshold variable. The third section conducts empirical analysis by first analyzing the current status quo of the level of ecological resilience of China's current resource-based cities, then analyzing the regression results based on each model, and providing further explanations for the regression results. The fourth section is the conclusion of the article.

## 2. Literature review

The concept of "ecological resilience" originates from physics. Resilience means returning to the original state [1], and the concept was introduced into the field of ecology by Holling in 1973, who created the concept of ecological resilience. Ecological resilience is mainly used to measure the ability of a region to resist risks and to further improve the resistance, adaptability, and resilience of the regional ecological environment [2]. Research on urban ecological resilience began with the study of the concept of resilience: Berkes (1998) defined resilience as the resistance of a system to the impact of destroying the original homeostasis and the speed of restoring the original homeostasis [3]. Holling (2003) argued that resilience reflects not only the system's ability to recover when it is damaged but also its ability to absorb change when it is dysfunctional [4]. With the continuous development of the conceptual theory of resilience, the concept of urban resilience emerged. Campanella (2006) studied the urban recovery of the city of New Orleans after experiencing Hurricane Katrina, and he argued that urban resilience is centered on a smarter population group [5]. Jha et al. (2013) argued that urban resilience is

mainly embodied in infrastructure resilience, institutional resilience, economic resilience and social resilience [6]. Since the turn of the 21st century, ecological protection has become the topic of the new era, and ecological resilience has also become a hotly debated topic among scholars. Pickett et al. (2014) argued that ecological resilience focuses on the adaptive capacity of ecosystems in the face of exogenous shocks as well as the ability to change in interactions within the system. Additionally ecological resilience is able to better reflect the sustainable development of a region [7]. Hu et al. (2021) studied the factors influencing urban transportation ecological resilience. The results showed that governance capacity, market activities, technological innovation capacity, trade openness, and financial resources have a significant impact on urban transportation ecological resilience [8]. Fu et al. (2023) showed that the coordinated development of urban ecological resilience and green technological innovation can effectively address the environmental pollution caused by carbon haze [9].

Innovation is a necessary driving force for economic development, and the implementation of an innovation-driven development strategy is an inevitable choice in the context of the new normal. As the formulator, implementer and guide of development strategy, the government's preference for innovation activities reflects the development prospects of innovation-driven development strategy. The government's financial expenditure on innovation activities is also the most effective way of allocating resources for innovation-driven development strategy (Lee, 2011 [10]). Therefore, in recent years, many scholars have conducted a series of studies on government innovation preferences. Some scholars have explored the factors influencing government innovation preferences: Zhang and Zou (1998) found that government innovation preferences will be inhibited by an increase in fiscal decentralization, which will further make the government and the market lose the "right to control" innovation activities, resulting in the continuous decline in innovation efficiency [11]. Bloom et al. (2016) argued that fierce competition in the international market will encourage the government to increase its investment in innovation, resulting in a sense of innovation preference [12]. Jiang (2020) found that the government innovation preferences will be affected by the degree of institutional perfection. Perfect and tight institutional conditions can effectively divide the relationship between the government and the market and avoid self-interested investment and the biased behavior of the government [13]. Some scholars have also carried out relevant research on the impact of government innovation preferences on innovation activities: Kleer (2010) argued that government innovation preferences will give the market a trend direction of industrial development, further leading the capital market to constantly engage in innovation activities [14]. Acemoglu et al. (2018) found that investment in innovation activities formed by government innovation preferences may have a crowding-out effect on private investment in innovation, which may lead to a more favorable environment for the government and result in a continuous loss of innovation efficiency [15]. Hu (2021) argued that government innovation preferences will increase regulations on cleaner production and punishments for polluting the environment for enterprises to promote technological innovation to improve the production efficiency of enterprises and to reduce the emission of pollutants [16]. Ghazala Aziz et al. (2023) and Cui et al. (2023) studied five East Asian countries, including China and Japan, as well as Gulf Cooperation Council countries, and they found that environmental technological innovations reduce the ecological footprint [17, 18]. Ye et al. (2024) found that governmental innovation preferences can significantly contribute to the development of the digital economy [19].

The essence of environmental decentralization is the mechanism for the division of financial and administrative rights for environmental protection, which is mainly reflected in the distribution of the right to speak and the division of responsibilities between the

central government level and the local government level for matters related to environmental governance. The aim is to realize the mutual incentives and tolerance of the central and local governments for environmental financial and administrative rights to maximally improve the efficiency of environmental governance (Peng Xing, 2016 [20]). Many scholars have conducted a series of studies on environmental decentralization. Among them, some scholars have discussed the advantages and disadvantages of environmental decentralization and centralization: Millimet (2003) argued that there are differences in the geographic environment, economic base and climatic conditions of different regions, that local governments have the advantage of access to information, and that environmental decentralization can enable local governments to carry out environmental governance activities more efficiently [21]. Comparing resource-based and non-resource-based cities in China, Zheng et al. (2023) found that environmental decentralization promotes industrial structure upgrading, which can further improve local environmental governance [22]. However, Fan (2009) argued that while decentralization can lead to more targeted environmental governance policies, environmental governance will bear a higher burden due to bribery as the number of regulatory agencies increases [23]. Li et al. (2021) argued that environmental decentralization can limit emissions at borders, strengthen the incentives for local governments to behave in an environmentally friendly manner, and most importantly, minimize corporate "free-riding" behavior [24]. Xu et al. (2023) found that the decentralization of centralized environmental regulation and environmental supervision promotes carbon emissions and pollutes the environment [25]. Other scholars have conducted a series of studies on the indicator measures of environmental decentralization. These scholars include Sigman (2007), who first constructed an indicator of environmental decentralization and argued that this indicator is affected by a number of factors, such as the variability of regulatory agencies in different countries, the variability of laws for decentralization and actual decentralization [26]. Fredriksson et al. (2014) used dummy variables for whether a country has a federal constitution and whether the governmental hierarchy is above four levels to measure environmental decentralization [27]. In contrast, Wu et al. (2020) and Fang et al. (2022) measured environmental decentralization using the degree of decentralization between the share of central environmental governors and the share of local environmental governors [28, 29].

   In summary, the literature has carried out a large number of studies on ecological resilience, government innovation preferences and environmental decentralization, forming a relatively perfect theoretical system and laying a more adequate theoretical foundation for this paper. However, at the same time, there are the following shortcomings: (1) Most of the literature is directly studies the single influencing factor of ecological toughness, and research on the mediating effect mechanism and threshold effect mechanism of ecological toughness is lacking. (2) Most of the literature focuses on the ecological resilience of the Pearl River Delta (PRD) region and the Yellow River Basin (YRB), while fewer relevant studies have focused on resource-based cities, which are a group of cities in urgent need of improving their ecological resilience to improve the status quo of resource extraction. On this basis, this study uses the educational level as a mediating variable and environmental decentralization as a threshold variable to investigate the mechanism of the impact of government innovation preferences on the ecological resilience of new resource-based cities. This study includes the following innovations: (1) It explores the path of ecological resilience from the perspective of government innovation preferences. (2) It introduces environmental decentralization as a threshold variable to explore the necessary conditions for government innovation preferences to have an impact on the ecological resilience of resource-based cities. (3) The selection of resource-based cities as the research object makes the conclusions of this paper more targeted.

## 3. Research hypotheses and model construction

### 3.1. Research hypotheses

Government innovation preferences can directly reflect the degree of importance attached to science and technology innovation activities through social development trends. For resource-based cities, the pillar industries will depend on the uncontrolled exploitation of resources and the constant impact on the ecological environment. This development mode will definitely lead to the "resource curse" phenomenon. Therefore, to alleviate this market paradox, the government can take the initiative to increase subsidies for science and technology innovation activities, which will not only increase the accumulation of local innovation capital in resource-based cities but also attract an influx of innovative talent from neighboring places to carry out more cutting-edge science and technology innovation activities. With the dual-factor guarantee of innovation talent and innovation capital, coupled with government innovation preferences for enterprises with stronger independent innovation ability (Guo Bing and Luo Shougui 2015 [30]), enterprises will have the willingness to carry out innovative activities to improve their production efficiency and sustainable resource extraction, which can further enhance the level of ecological resilience of resource-based cities. At the same time, the government will inevitably increase the confidence of government investment in science and technology innovation in the face of the production vitality shown by enterprises and the increasingly improved ecological environment, thus forming a virtuous circle (Liu Feiran and Hu Lijun 2020 [31]). However, excessive government innovation preferences may cause the government's financial expenditure on science and technology to crowd out the due cost of environmental governance, thus forming an imbalance in which the practical activities of environmental governance are unable to meet the requirements of the level of technological innovation, thus resulting in a decline in the level of ecological resilience of resource-based cities. Therefore, based on the theoretical research above, the following hypothesis is proposed:

H1: Appropriate government innovation preferences can enhance the level of ecological resilience of resource-based cities, and excessive government innovation preferences can cause a decline in the level of ecological resilience of these cities.

Improving the educational level is one of the important fundamentals for realizing the concentration of innovative talent. Such improvement can promote an improvement in social labor efficiency and productivity (EASTERLY 1993 [32]). The main purpose of the government's investment in innovation is to attract the agglomeration of two innovation factors, namely, innovation manpower and innovation capital. The agglomeration of innovative manpower plays an important role in enhancing the innovation capacity of a region (TEIXEIRA, 2004 [33]). It can attract an influx of talent, thus forming an agglomeration effect through innovation capital and the innovation practice base. However, compared with an influx of external innovation talent, actively improving the educational level and cultivating the science and technology innovation talent in a region are more beneficial for the adaptability of innovation activities and the sustainability of such activities. Therefore, one of the important means for government innovation preferences to enhance the ecological resilience of resource-based cities is to actively carry out educational activities and improve the educational level in the region, which will thus lead to the execution of innovative practical activities, an improvement in the means of environmental governance, and an increase in the level of ecological resilience. Therefore, based on the theoretical research above, the following hypothesis is proposed:

H2: The educational level has a mediating effect on the mechanism of the impact of government innovation preferences on the ecological resilience of resource-based cities.

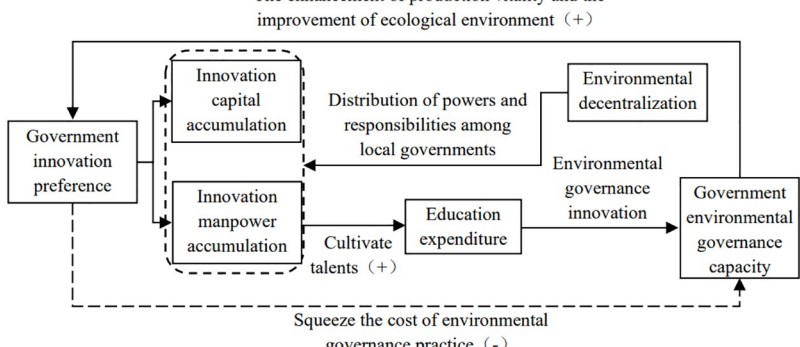

**Fig 1. Mechanisms of government innovation preferences on ecological resilience in resource-based cities.**

The construction of the environmental decentralization system forms the basis for measuring the rights and responsibilities of the central and local governments for environmental governance, and only by clarifying the relationship between the rights and responsibilities of these two levels of government in ecological protection and avoiding the "one-size-fits-all" decentralization of environmental governance can initiatives related to ecological civilization construction be optimized (LOVO 2018; Xu Pei and Wu Shanshan 2020 [34, 35]). Therefore, environmental decentralization constitutes an important guarantee for the central government to realize environmental governance work based on local governments. Additionally, appropriate environmental decentralization can effectively alleviate the pressure on local governments in terms of facing the input of science and technology innovation-related activities to utilize the results of innovations and hence apply them to environmental governance work, thereby further enhancing the level of ecological resilience of resource-based cities. Therefore, based on the theoretical research above, the following hypothesis is proposed:

H3: Environmental decentralization produces a threshold effect in the mechanism of the impact of government innovation preferences on the ecological resilience of resource-based cities.

Based on the analysis of the theoretical mechanism of the impact of government innovation preferences on the ecological resilience of resource-based cities above, the diagram of the action mechanism can finally be derived, as shown in Fig 1.

## 3.2. Model construction

### 3.2.1. The entropy value method: A measure of the ecological resilience of resource-based cities.
The ecological resilience of resource-based cities reflects the ability of the ecosystem to evolve from disorder to order when subjected to shocks. When resource-based cities are subjected to specific production activities, such as resource extraction and processing emissions, the orderly functioning of the ecosystem will be subjected to greater impacts, thus increasing the risk of ecosystem destruction. To ensure the normal operation of the ecosystem, it is possible to improve the level of the following three subsystems: ecosystem resistance, adaptive capacity and restorative capacity. Therefore, this study establishes an ecological resilience evaluation index for resource-based cities based on the principles of systematic science, indicator hierarchy, and data availability. Additionally, based on the results of the studies conducted by Wu et al. (2020) and Tang (2023) [36, 37], this study establishes a set of 10 indicators for the

**Table 1. Indicator system for evaluating the ecological resilience in resource-based cities.**

| Target | Subsystems | Standard Floor | Efficacy |
|---|---|---|---|
| Ecological resilience in Resource-based cities | Resistance capacity | PM2.5 (mg/m$^3$) | - |
| | | Sulfur dioxide emissions (10000 tons) | - |
| | | Industrial wastewater discharge (10000 tons) | - |
| | | Non-hazardous treatment rate of domestic waste (%) | + |
| | Adaptive capacity | Value added of the secondary sector as a share of GDP (%) | - |
| | | Water consumption in residential households (10000 tons) | - |
| | Recuperative capacity | Area of green space (hectares) | + |
| | | Number of vehicles in operation at the end of the year (units) | + |
| | | Centralized treatment rate of sewage treatment plants (%) | + |
| | | Acquisition of green patents (pieces) | + |

three subsystems, i.e., ecosystem resistance, adaptive capacity, and restorative capacity. The ecological resilience evaluation index system for resource-based cities is shown in Table 1. Considering that the entropy value method reduces the interference of subjective factors when assigning values and can accurately calculate the relevant comprehensive index of each region, it is used to measure the green development index system. Since the entropy value method is a common method of index assignment and the algorithm has too much content, it will not be repeated here. For the specific content of the model, see Delgado et al. (2016), Yan et al. (2021) and related studies [38, 39].

**3.2.2. Measurement of environmental decentralization.** Based on the specificity of China's administration and the reasonableness and availability of data, this paper draws on the studies of Wu et al. (2020) and Fang et al. (2022) and measures the degree of environmental decentralization (*END*) on the basis of the distribution of urban unit employment in the water conservancy, environment and public facilities management industry in each city [28, 29]. To avoid the problem of endogeneity between environmental decentralization and economic development, the economic deflation factor is incorporated into the formula of the measurement of environmental decentralization. The method of measuring environmental decentralization is shown in Eq (1):

$$END_{it} = \frac{LE_{it}/LP_{it}}{GE_t/GP_t}[1 - (GDP_{it}/GDP_t)] \tag{1}$$

where *i* and *t* represent cities and years, respectively. $LE_{it}$ denotes the number of persons employed in urban units in the water conservancy, environment and public facilities management industry in prefecture-level cities; $LP_{it}$ denotes the total population at the end of the year in prefecture-level cities; and $GE_t$ denotes the number of persons employed in urban units in the water conservancy, environment and public facilities management industry in the $t_{th}$ year of China $GP_t$ denotes the total population at the end of the year in the yth year of China. [1 − ($GDP_{it}/GDP_t$)] is an economic scaling factor.

**3.2.3. Benchmark regression and mediating effects models.** Innovation-driven development strategy is an important initiative for China to realize the optimization and upgrading of its industrial structure. Additionally, the development of innovation activities in resource-based cities can effectively alleviate the environmental pressure brought by resource development and ensure that the resistance and resilience of ecological resilience is always maintained at a high level. Therefore, to explore whether government innovation preferences will have direct effects and marginal beneficial effects on ecological resilience in resource-based cities

and to verify H1, this study first establishes a baseline regression model by taking the primary and secondary terms of government innovation preferences as explanatory variables. Given that the government's regional innovation activities are mainly generated through its fiscal expenditure behavior [10], the amount of local fiscal science and technology expenditure in each region is selected to measure the level of government innovation preferences. In addition, to test H2, this paper introduces the educational level as a mediating effect, and the ratio of education expenditure to GDP is selected to measure the educational level to explore whether the educational level plays a mediating role in government innovation preferences in the ecological resilience of resource-based cities. The specific formulation of the model is shown in Eqs (2)–(5):

$$ECR_{i,t} = \alpha_0 + \beta_1 INP_{i,t} + \beta_2 INP_{i,t}^2 + \mu_i + \sigma_t + \varepsilon_{i,t} \tag{2}$$

$$ECR_{i,t} = \alpha_0 + \beta_1 INP_{i,t} + \beta_2 INP_{i,t}^2 + \sum_{j=1}^{j} \lambda_j X_{i,t,j} + \mu_i + \sigma_t + \varepsilon_{i,t} \tag{3}$$

$$EDL_{i,t} = \alpha_0 + \alpha_1 INP_{i,t} + \alpha INP_{i,t}^2 + \sum_{j=1}^{j} \lambda_j X_{i,t,j} + \mu_i + \sigma_t + \varepsilon_{i,t} \tag{4}$$

$$ECR_{i,t} = \alpha_0 + \alpha_1 INP_{i,t} + \alpha_2 INP_{i,t}^2 + \alpha_3 EDL_{i,t} + \sum_{j=1}^{j} \lambda_j X_{i,t,j} + \mu_i + \sigma_t + \varepsilon_{i,t} \tag{5}$$

Eqs (2) and (3) are the baseline regression models of the effect of government innovation preferences on the ecological resilience of resource-based cities without and with control variables added, respectively. [$ECR_{i,t}$] denotes the ecological resilience composite score of the $i_{th}$ city in the $t_{th}$ year, and $INP_{i,t}$ denoted the level of government innovation preferences of the $i_{th}$ city in the $t_{ht}$ year. Eqs (4) and (5) are the models of the mediating effect of the educational level on the impact of government innovation preferences on the ecological resilience of resource-based cities without and with control variables added, respectively. $EDL_{i,t}$ $EDL_{i,t}$ denotes the educational level of the $i_{th}$ city in the $t_{th}$ year, and $X_{i,t,j}$ denotes $j_{th}$ control variable of the $i_{th}$ city in the $t_{th}$ year. $E_{i,t}$ represents the random error term.

To eliminate the interference of external factors in the mechanism of the impact of government innovation preferences on the ecological resilience of resource-based cities and to make the model regression results more accurate and reliable, this study adds the following control variables to the regression model:

1. Level of economic development (*PGDP*). The level of economic development in resource-based cities often relies on the destruction of the ecological environment to carry out resource extraction, and in the long run, economic growth will lead to climate anomalies and thus destroy the ecological structure (Gulzara et al., 2022 [40]). However, with the strengthening of environmental awareness and the industrial transformation and upgrading in resource-based regions in recent years, the coordinated development of the level of economic development and the ecological environment has achieved better results. This study uses the per capita GDP of each city to measure the level of economic development.

2. Government intervention (*GOV*). The main purpose of government intervention is to rationalize resource allocation and compensate for the weakening of ecosystem resilience caused by market failure. However, excessive government intervention may also lead to a

decline in the efficiency of environmental governance-related initiatives, resulting in a decline in the adaptive capacity of ecosystems. In this paper, the proportion of fiscal expenditure in GDP is chosen to measure government intervention.

3. Collaborative industrial agglomeration (*COG*). Industrial synergistic agglomeration can promote the learning effect, cost effect, and competition effect among industries, thus enhancing the optimization of the industrial structure, improving production efficiency, and further improving the level of ecological resilience. This paper first draws on the research of Wu et al. (2022) [41], adopting the location entropy method to measure the level of industrial agglomeration of the manufacturing industry and the industrial agglomeration of the productive service industry. Then, to further reflect the synergistic height and synergistic quality between industries at the same time, this study draws on the research results of Ellison (1997, 2010 [42, 43]), adopting the E-G index to measure collaborative industrial agglomeration.

4. Level of financial development (*FIA*). The level of financial development reflects the size of the capital market in resource-based cities, and a larger capital market size can form capital accumulation, and further investment in ecological management will increase. In this paper, the level of financial development is measured as the proportion of the balance of the RMB loans of financial institutions to the regional GDP at the end of the year.

**3.2.4. Threshold effects model.** The level of environmental decentralization reflects the degree of balance in the division of power and responsibility between the central government and local governments for environmental governance. Additionally, appropriate decentralization can reduce the cost of environmental regulation by local governments, alleviate resource constraints, and promote green production and environmental protection (Wu et al., 2020 [28]; Sigman, 2014 [44]). However, excessive decentralization may result in local governments being unable to implement the environmental governance standards set by the central government in full compliance, causing local governments to attract capital inflows to protect the competitive advantages of local firms, leading to a decrease in the incentives for firms to reduce pollution, and ultimately weakening the quality of the environment (Lin and Xu, 2022 [45]; Ren et al., 2023 [46]). Therefore, the positive impact of government innovation preferences on the ecological resilience of resource-based cities is predicated on the premise that local governments have a certain degree of voice in the environmental governance of their regions and that they must ensure that they can fulfill the environmental protection goals set by the central government with guaranteed quality. Therefore, to further explore whether environmental decentralization affects the mechanism of the impact of government innovation preferences on the ecological resilience of resource-based cities and to verify H3, we introduce the threshold variable of environmental decentralization and construct a threshold effect model to investigate the differences in the mechanism of the impact of government innovation preferences on the ecological resilience of resource-based cities under different levels of environmental decentralization.

$$
\begin{aligned}
ECR_{i,t} = {} & \alpha_0 + \alpha_1 INP_{i,t} I(END \le r) + \alpha_2 INP_{i,t} I(END > r) + \alpha_3 INP_{i,t}{}^2 I(END \le r) \\
& + \alpha_4 INP_{i,t}{}^2 I(END > r) + \alpha_n X_{i,t} + \mu_i + \sigma_t + \varepsilon_{i,t}
\end{aligned}
\tag{6}
$$

In Eq (6), *END* is the environmental weighting of the threshold variable (see 3.2.2 Measurement of environmental decentralization for specific measures), *r* is the threshold value, and the remaining variables carry the same meaning as above.

**Table 2. Distribution of study areas.**

| city | P | L | S | city | P | L | S | city | P | L | S | city | P | L | S |
|---|---|---|---|---|---|---|---|---|---|---|---|---|---|---|---|
| Anshan | 4 | 1 | 1 | Hegang | 3 | 2 | 2 | Luzhou | 3 | 3 | 1 | Tongchuan | 3 | 3 | 3 |
| Anshun | 2 | 3 | 2 | Heihe | 2 | 2 | 3 | Lvliang | 2 | 2 | 3 | Tonghua | 4 | 2 | 3 |
| Baise | 2 | 1 | 3 | Hengyang | 2 | 2 | 1 | Maanshan | 4 | 2 | 2 | Tongling | 3 | 2 | 2 |
| Baishan | 3 | 2 | 3 | Hezhou | 1 | 1 | 3 | Mudanjiang | 2 | 2 | 2 | Weinan | 2 | 3 | 2 |
| Baiyin | 3 | 3 | 3 | Huaibei | 3 | 2 | 2 | Nanchong | 1 | 3 | 1 | Wuhai | 3 | 2 | 2 |
| Baoji | 2 | 3 | 2 | Huainan | 2 | 2 | 1 | Nanping | 2 | 1 | 3 | Wuwei | 1 | 3 | 3 |
| Baoshan | 2 | 3 | 3 | Huangshi | 3 | 2 | 2 | Nanyang | 4 | 2 | 2 | Xianyang | 1 | 3 | 1 |
| Baotou | 4 | 2 | 1 | Huludao | 4 | 1 | 2 | Ordos | 1 | 2 | 2 | Xingtai | 2 | 1 | 1 |
| Benxi | 2 | 1 | 2 | Hulunbuir | 1 | 2 | 3 | Panjin | 4 | 1 | 2 | Xinyu | 3 | 2 | 2 |
| Bozhou | 2 | 2 | 2 | Huzhou | 2 | 1 | 2 | Panzhihua | 2 | 3 | 2 | Xinzhou | 2 | 2 | 3 |
| Changzhi | 2 | 2 | 1 | Jiaozuo | 3 | 2 | 2 | Pingdingshan | 2 | 2 | 2 | Xuancheng | 2 | 2 | 3 |
| Chende | 2 | 1 | 2 | Jilin | 2 | 2 | 1 | Pingliang | 2 | 3 | 3 | Xuzhou | 4 | 1 | 1 |
| Chenzhou | 2 | 2 | 2 | Jinchang | 2 | 3 | 3 | Pingxiang | 3 | 2 | 2 | Ya'an | 2 | 3 | 3 |
| Chifeng | 2 | 2 | 1 | Jincheng | 2 | 2 | 2 | Puyang | 3 | 2 | 2 | Yan'an | 1 | 3 | 3 |
| Chizhou | 2 | 2 | 3 | Jingdezhen | 3 | 2 | 2 | Qingyang | 1 | 3 | 3 | Yangquan | 2 | 2 | 2 |
| Chuzhou | 2 | 2 | 3 | Jining | 2 | 1 | 1 | Qitaihe | 3 | 2 | 3 | Yichun(HLJ) | 3 | 2 | 3 |
| Daqing | 2 | 2 | 1 | Jinzhong | 2 | 2 | 2 | Qujing | 2 | 3 | 2 | Yichun(JX) | 2 | 2 | 2 |
| Datong | 2 | 2 | 1 | Jixi | 2 | 2 | 2 | Sanmenxia | 2 | 2 | 3 | Yulin | 1 | 3 | 2 |
| Dazhou | 2 | 3 | 2 | Karamay | 2 | 3 | 3 | Sanming | 2 | 1 | 3 | Yuncheng | 2 | 2 | 2 |
| Dongying | 2 | 1 | 2 | Liaoyuan | 3 | 2 | 3 | Shaoguan | 3 | 1 | 2 | Yunfu | 2 | 1 | 3 |
| Ezhou | 2 | 2 | 3 | Lijiang | 4 | 3 | 3 | Shaoyang | 2 | 2 | 2 | Zaozhuang | 3 | 1 | 1 |
| Fushun | 3 | 1 | 1 | Lincang | 2 | 3 | 3 | Shizuishan | 3 | 3 | 3 | Zhangjiakou | 2 | 1 | 1 |
| Fuxin | 3 | 1 | 2 | Linfenn | 2 | 2 | 2 | Shuangyashan | 3 | 2 | 3 | Zhangye | 4 | 3 | 3 |
| Ganzhou | 2 | 2 | 1 | Linyi | 4 | 1 | 1 | Shuozhou | 1 | 2 | 3 | Zhaotong | 1 | 3 | 3 |
| Guang'an | 2 | 3 | 3 | Liupanshui | 1 | 3 | 2 | Songyuan | 1 | 2 | 3 | Zibo | 4 | 1 | 1 |
| Guangyuan | 2 | 3 | 3 | Longnan | 1 | 3 | 3 | Suqian | 4 | 1 | 2 | Zigong | 2 | 3 | 2 |
| Handan | 2 | 1 | 1 | Longyan | 2 | 1 | 2 | Suzhou | 2 | 2 | 2 | | | | |
| Hebi | 2 | 2 | 2 | Loudi | 2 | 2 | 2 | Tai'an | 2 | 1 | 1 | | | | |
| Hechi | 2 | 1 | 3 | Luoyang | 4 | 2 | 1 | Tangshan | 4 | 1 | 1 | | | | |

Note: P represents the stage of development in which the resource city is located: 1 for growing cities; 2 for mature cities; 3 for regenerating cities; and 4 for declining cities. L represents the region in which the resource city is located: 1 for cities in the east; 2 for cities in the central part of the country; and 3 for cities in the western part of the country. S represents the size of the city: 1 for large cities; 2 for medium-sized cities; and 3 for small cities.

## 4. Empirical analysis

### 4.1. Sample selection and data sources

In In this study, 2009–2020 data covering 113 resource-based cities in China are selected, and the regional distribution of the study is shown in Table 2. The data for each indicator are obtained from the 2010–2021 China Statistical Yearbook, China Environmental Statistical Yearbook, and Compendium of Statistical Data for the 60 Years of New China, as well as the Statistical Yearbook and Statistical Bulletin of National Economic and Social Development of each prefecture-level city in China. Given that there are some missing data, this paper uses linear interpolation to add the missing data. The descriptive statistics of each variable are shown in Table 3.

**Table 3. Descriptive statistics of variables.**

| Variable | Obs | Mean | Std. Dev. | Min | Max |
|---|---|---|---|---|---|
| ECR | 1,356 | 0.2897 | 0.0989 | 0.1394 | 0.7406 |
| INP | 1,356 | 2.8264 | 3.7108 | 0.0753 | 29.612 |
| EDL | 1,356 | 0.1753 | 0.039 | 0.0357 | 0.2918 |
| END | 1,356 | 1.0918 | 0.7195 | 0.0230 | 4.6938 |
| PGDP | 1,356 | 4.4225 | 3.0229 | 0.0099 | 25.6877 |
| FID | 1,356 | 0.0096 | 0.005 | 0.0014 | 0.0295 |
| SIA | 1,356 | 2.2139 | 0.9547 | 0.6414 | 20.2025 |
| FIL | 1,356 | 1062.828 | 1007.925 | 79.4027 | 8953 |

## 4.2. Analysis of the current situation of the ecological resilience of resource-based cities

The level of ecological resilience reflects the importance that a region places on ecological and environmental protection, and the establishment of a more resilient infrastructure can effectively promote the continuation of innovative practices and industries to achieve the goal of sustainable development (Bless et al., 2022 [47]). To more directly analyze and study the current status of the ecological resilience of resource-based cities, this study takes 10 indicators of the three ecological resilience subsystems of resource-based cities, namely, resistance, adaptation and recovery, as the basis for measurement, and it adopts the entropy value method to calculate the ecological resilience composite scores of the 113 sample cities. First, kernel density maps of the 113 resource-based cities from 2009 to 2020 are drawn with the help of MATLAB software to analyze the dynamic evolutionary characteristics of the ecological resilience of China's resource-based cities, as shown in Fig 2. Then, the calculation results of four years, i.e., 2009, 2013, 2017, and 2020, were further selected and bar graphs were drawn with the help of Excel software to analyze the development status of the ecological resilience of the 113 resource-based cities from the three perspectives of time, space and development history. The results are shown in Fig 3.

Fig 2 shows that, first, from the perspective of the wave crest distribution, the comprehensive scores of ecological resilience of resource-based cities in China are mostly concentrated in the range of 0.1–0.3. This result indicates that the current level of ecological resilience of resource-based cities in China is low and that it is necessary to further strengthen the requirements of ecological civilization construction to improve the level of ecological resilience of resource-based cities. Second, from the direction of movement of the wave crest, the center of the overall density function moves to the right, but the amplitude of the moving trend first increases and then decreases. These results indicate that the ecological resilience of China's resource-based cities is constantly improving over the whole 2009–2020 period. In the process of improvement, at the early stage, given the restriction and guidance of relevant ecological civilization governance policies in China, the improvement speed of the ecological resilience of resource-based cities gradually accelerated. However, as the ecological civilization undertakings in various cities continually deepen, the heterogeneity of the resource development stage, the difference in capital accumulation, and other influencing factors are reflected, and the marginal benefits of relevant policies continue to decline. Due to the lack of long-term effective governance effects, the speed of ecological resilience improvement slows down. Finally, from the perspective of the changing trend of the wave crest width, the width of the wave crest is increasing, which reflects that the difference in the ecological resilience of resource-based cities

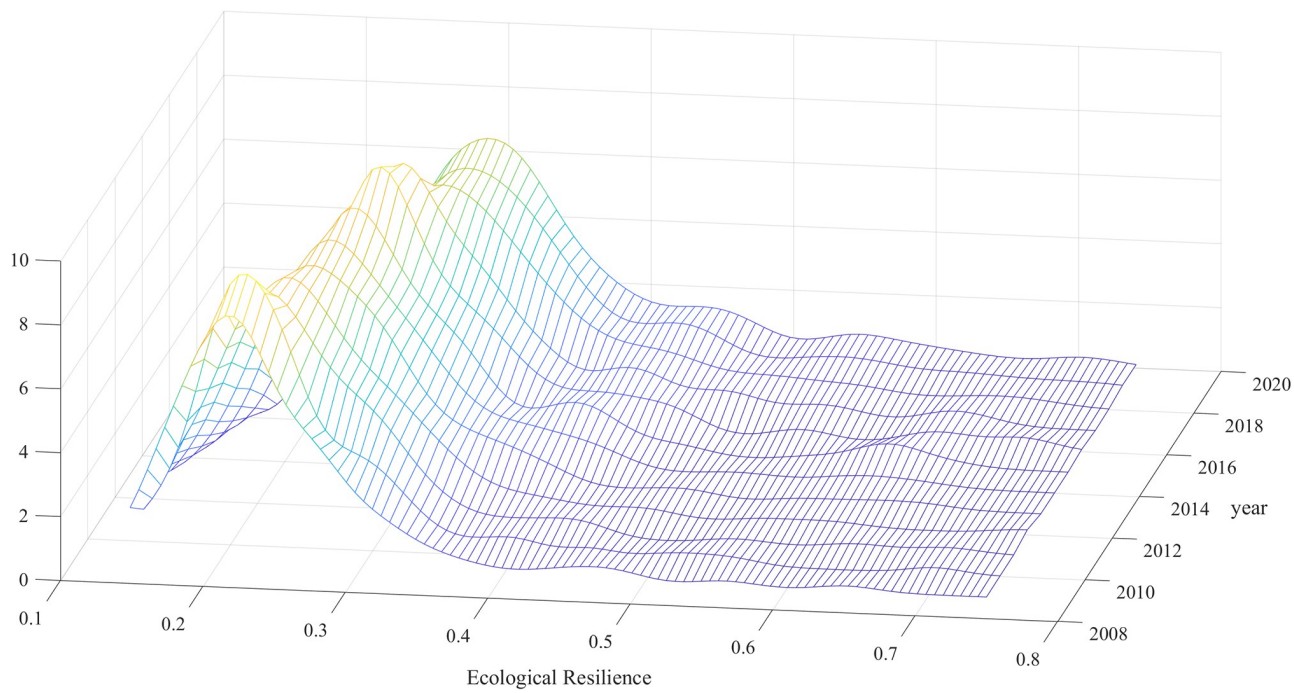

**Fig 2. Dynamic evolution of ecological resilience in China's resource-based cities, 2009–2020.**

in China is gradually increasing. The reason for this phenomenon is that different resource-based cities have strong differences in terms of their geographical location, development stage, and initial capital accumulation. Therefore, when the government guides resource-based cities to improve measures related to ecological civilization construction, there are large differences in the execution power and adaptability of each city, resulting in a large difference in the ecological resilience of China's resource-based cities.

As shown in Fig 3, from the perspective of spatial analysis, the comprehensive scores of 96 of the 113 sample resource-based cities in 2009 were within the range of 0.1–0.3. This result indicates that the ecological resilience of most resource-based cities in China was at a low level at the beginning of the study period. However, the comprehensive scores of six central and eastern cities represented by Tangshan (0.43) and Xuzhou (0.49) exceeded 0.4 points, indicating that the level of ecological resilience of Central and Eastern China significantly differed from that of Western China at the beginning of the study period. In terms of the spatial distribution of ecological resilience scores in 2013, most resource-based cities were no longer in the lowest score range (0.1–0.2), but 14 sample cities in Western and Central China, including Longnan (0.20), Lvliang (0.21) and Hechi (0.23), were still in the lowest range, and the development of ecological resilience stagnated. These results further demonstrate the unbalanced development of China's ecological civilization construction. In 2017 and 2020, the distribution of the level of China's ecological resilience showed a positive trend. In particular, by 2020, all sample cities were no longer in the score range of the lowest level of ecological resilience, but most cities were still in the middle level of 0.2–0.4. Only 20 and 22 cities scored more than 0.4 on ecological resilience at these two time points.

From the analysis of the time dimension, it can be concluded that in the 2009–2013 period, the comprehensive scores of ecological resilience of 98 of the 113 sample cities improved, but there was also a decline in the comprehensive scores of ecological resilience of 15 cities,

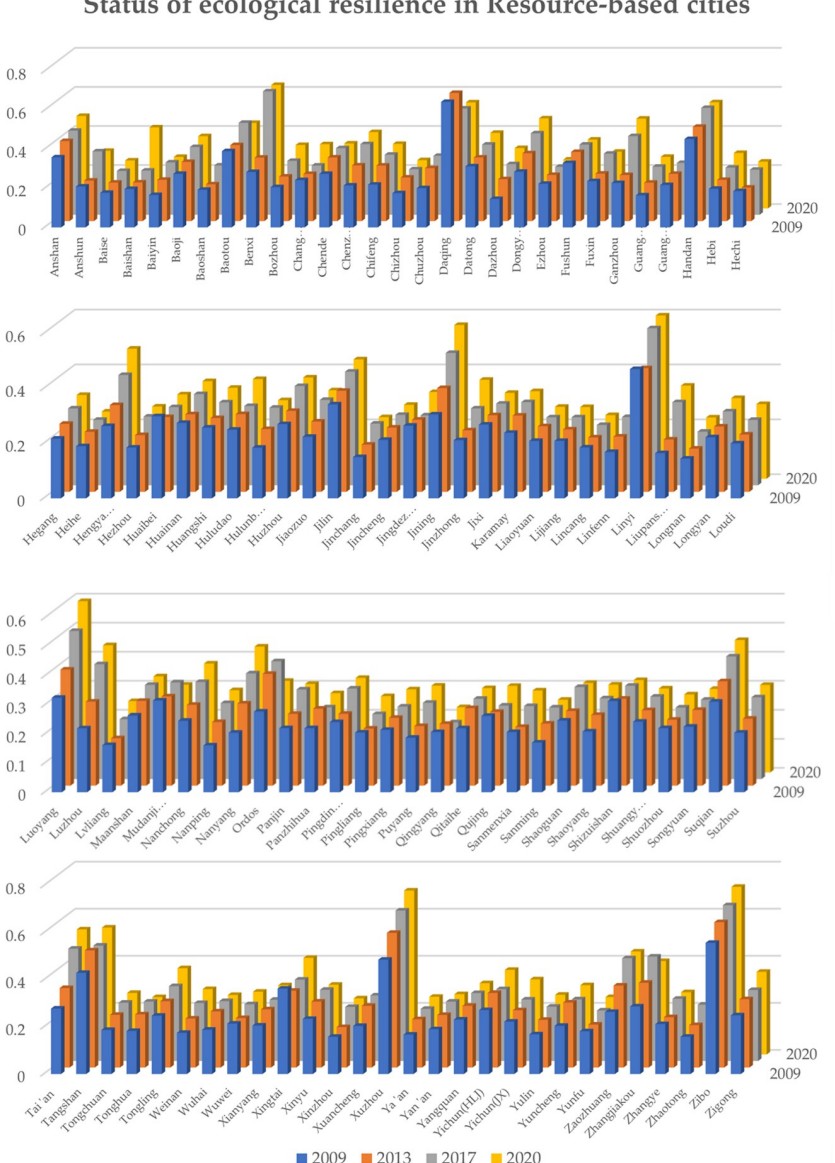

**Fig 3. Status of ecological resilience in resource-based cities.**

represented by Xingtai and Huaibei. These results show that the ecological resilience of Chinese resource-based cities at the beginning of the study period had an overall good development trend. However, in the 2013–2017 period, the number of sample cities with improved ecological resilience reached 105, and only 8 cities, including Daqing and Panzhihua, experienced a decline in their composite scores of ecological resilience. These results further demonstrate the improvement in the development of ecological resilience in China's resource-based cities. This suggests that after the Chinese State Council officially released the "Sustainable Development Plan for China's Resource-Based Cities (2013–2020)", most of China's resource-based cities responded positively, but some western and northern cities may have responded insufficiently to the policy due to their economic volume, infrastructure, and inherited

development patterns. However, in the latter part of the study period, from 2017 to 2020, the scores of cities were not only much lower than those of the 2013–2017 period but also lower than those of the 2009–2013 period, although there were still 95 cities with improved composite scores of ecological resilience. These results fully prove that with the advancement of the "Sustainable Development Plan for China's Resource-Based Cities (2013–2020)", the marginal benefits of this policy for resource-based cities in terms of completing industrial transformation, achieving harmonious coexistence between humans and nature, and completing the qualitative leap in the cause of ecological civilization construction have been declining. In this context, the results also show the need for China to launch the "14th Five-Year Plan for Promoting the High-Quality Development of Resource-Based Areas" in 2021 to realize a new development path for resource-based cities.

Analyzing the overall development history, 111 of the 113 cities in this study improved their ecological resilience scores in 2020 compared to 2009. This result reflects the effectiveness of China's measures to realize the goal of changing resource-based cities' status quo of uncontrolled resource extraction and moving away from the "ecological curse". However, Daqing city and Mudanjiang city in Heilongjiang Province showed a downward trend in ecological resilience, requiring timely intervention by the government to improve the mechanism of measures related to ecological civilization construction in the region to achieve full coverage of ecological resilience improvement in resource-based cities. Overall, the ecological resilience of China's resource-based cities shows a good upward trend. However, the decrease in the marginal benefits of governmental measures constitutes a warning that Chinese authorities should be ready to adjust and introduce better regulations and policies related to the ecological civilization construction of resource-based cities based on different stages of development and the current social situation.

## 4.3. Benchmark regression and mediating effect analysis

**4.3.1. Benchmark regression.**　To explore the mechanism of the direct effect of government innovation preferences on the ecological resilience of resource-based cities, this study calculates the benchmark regression results with Stata software. The results are shown in Table 4 (1)-(2), where (1) and (2) represent the results of the influence of government innovation preferences on the ecological resilience of resource-based cities before and after adding control variables, respectively. The results show that the primary terms (0.01619*** and 0.00510**) of government innovation preferences have a significant positive effect on the ecological resilience of resource-based cities regardless of whether control variables are added, and the secondary terms (-0.00036*** and -0.00013**) have a significant negative effect on the ecological resilience of resource-based cities. The reason for these results is that the government's investment in innovative activities can effectively enhance the level of ecological resilience in resource-based cities, but excessive innovation preferences may have a crowding-out effect on other costs of ecological governance, resulting in a decline in the level of ecological resilience. Regarding the control variables, the level of economic development (0.00901***), collaborative industrial agglomeration (0.00357**), and the level of financial development (0.00003***) have a significant positive impact on the ecological resilience of resource-based cities, but the coefficient of the impact of fiscal decentralization is negative and not significant.

**4.3.2. Mediation effects analysis.**　To further explore the mechanism of the impact of government innovation preferences on the ecological resilience of resource-based cities, this paper selects the educational level(*EDL*) as a mediating variable to establish a model of the mediating effect of government innovation preferences on the ecological resilience of

**Table 4. Benchmark regression results and mediation effect results.**

| Variables | (1) | (2) | (3) | (4) |
|---|---|---|---|---|
| INP | 0.01619*** | 0.00510** | 0.00278** | 0.00574*** |
|  | (7.19) | (2.48) | (2.42) | (2.76) |
| INP_2 | -0.00036*** | -0.00013** | -0.00010* | -0.00015** |
|  | (-3.59) | (-2.03) | (-1.83) | (-2.24) |
| EDL |  |  |  | -0.23391*** |
|  |  |  |  | (-3.10) |
| PGDP |  | 0.00901*** | -0.00166 | 0.00862*** |
|  |  | (4.86) | (-1.64) | (4.76) |
| GOV |  | -1.17673 | -2.56756*** | -1.77731 |
|  |  | (-0.96) | (-2.71) | (-1.36) |
| SIA |  | 0.00357** | -0.00129 | 0.00327** |
|  |  | (2.52) | (-1.19) | (2.13) |
| FIL |  | 0.00003*** | -0.00000 | 0.00003*** |
|  |  | (3.55) | (-1.46) | (3.56) |
| Constant | 0.25175*** | 0.21167*** | 0.20833*** | 0.26040*** |
|  | (55.42) | (19.50) | (27.31) | (12.06) |
| Obs | 1,356 | 1,356 | 1,356 | 1,356 |
| Number of id | 113 | 113 | 113 | 113 |
| R − squared | 0.247 | 0.452 | 0.073 | 0.464 |
| Company FE | YES | YES | YES | YES |
| Control variables | NO | YES | YES | YES |
| F test | 0 | 0 | 0.000249 | 0 |
| r2_a | 0.246 | 0.450 | 0.0689 | 0.461 |
| F | 44.96 | 46.60 | 4.731 | 50.21 |

Note:

*, **, and *** indicate significant at the 10%, 5%, and 1% levels, respectively; values in parentheses are t-values. Same as below.

resource-based cities. The results are calculated using Stata software, as shown in Table 4 (3) and 4 (4).

Table 4(3) shows the regression results of the impact of government innovation preferences on the educational level. The results show that the primary term of government innovation preferences has a significantly positive effect on the educational level at the 5% level (0.00278**), and the secondary term has a significantly negative effect at the 10% level (-0.00010*). Table 4 (4) shows the regression results of the impact of the educational level and government innovation preferences on the ecological resilience of resource-based cities. The primary item of government innovation preferences (0.00574***), the primary term of government innovation preferences (-0.00015**) and the educational level (-0.23391***) are all significant at least at the 5% level. The results show that the educational level plays a mediating role in the mechanism of the impact of government innovation preferences on the ecological resilience of resource-based cities.

**4.3.3. Heterogeneity analysis.** To further explore the heterogeneous role of government innovation preferences in the ecological resilience of resource-based cities, this subsection divides resource-based cities into cities in China's eastern, central, and western regions, divides them into large, medium-sized, and small cities based on city size, and conducts regression analyses of their impact mechanisms.

**Table 5. Results of regression analysis of regional heterogeneity.**

| Variables | (1) | (2) | (3) | (4) | (5) | (6) |
|---|---|---|---|---|---|---|
| INP | 0.01198** | -0.00238 | 0.01412*** | 0.00682*** | 0.03165*** | 0.01564*** |
| | (2.17) | (-0.51) | (6.15) | (2.88) | (7.22) | (3.04) |
| INP_2 | -0.00020 | 0.00005 | -0.00024*** | -0.00013* | -0.00107*** | -0.00053*** |
| | (-0.90) | (0.31) | (-2.98) | (-1.96) | (-7.35) | (-3.13) |
| PGDP | | 0.00712** | | 0.00883*** | | 0.00975** |
| | | (2.09) | | (3.67) | | (2.68) |
| GOV | | -4.57746 | | 0.61023 | | 1.28000 |
| | | (-1.50) | | (0.49) | | (0.85) |
| SIA | | 0.00471*** | | -0.01274 | | 0.00182 |
| | | (3.96) | | (-1.21) | | (0.41) |
| FIL | | 0.00004*** | | 0.00002* | | 0.00002 |
| | | (4.48) | | (1.85) | | (1.40) |
| Constant | 0.31478*** | 0.29776*** | 0.24604*** | 0.22839*** | 0.20117*** | 0.15681*** |
| | (21.39) | (9.05) | (46.82) | (10.07) | (34.83) | (8.88) |
| Obs | 336 | 336 | 660 | 660 | 360 | 360 |
| Number of id | 28 | 28 | 55 | 55 | 30 | 30 |
| R − squared | 0.134 | 0.472 | 0.353 | 0.478 | 0.397 | 0.598 |
| Company FE | YES | YES | YES | YES | YES | YES |
| Control variables | NO | YES | NO | YES | NO | YES |
| F test | 0.00015 | 0 | 0 | 0 | 0 | 0 |
| r2_a | 0.128 | 0.462 | 0.351 | 0.473 | 0.394 | 0.591 |
| F | 12.42 | 18.81 | 25.19 | 18.65 | 27.45 | 14.63 |

*(1) Regional heterogeneity*. The differences in the level of economic development, production and living styles, and the government's focus on resource-based cities in different regions will cause certain differences in government innovation preferences and ecological resilience. Therefore, this study divides resource-based cities into cities in China's eastern, central and western regions and conducts regression analysis on the impact mechanism of government innovation preferences on the ecological resilience of resource-based cities. The results are shown in Table 5: (1), (3), and (5) represent the regression results of eastern, central, and western resource-based cities without adding control variables, respectively and (2), (4), and (6) represent the regression results of eastern, central, and western resource-based cities after adding control variables, respectively.

As shown in Table 5, in the regression results with no control variables, the primary term of government innovation preferences has a positive effect on the ecological resilience of resource-based cities in the eastern (0.01198**), central (0.01412***) and western (0.03165***) regions. The coefficients of the effect are as follows (from highest to lowest): western, central and eastern cities. The effect of the quadratic term of government innovation preferences on the ecological resilience of resource-based cities in the eastern region is nonsignificant, the effect on resource-based cities in the central (-0.00024***) and western (-0.00107***) regions is negative. Additionally, the absolute value of the western region is larger than that of the central region. After adding control variables, government innovation preferences are not significant for the ecological resilience of resource-based cities in the eastern region. The coefficient of the primary term is higher in the western region (0.01564***) than in the central region (0.00682***), and the absolute value of the coefficient of the secondary term is higher in the western region (-0.00053***) than in the central region (-0.00013*). These results are basically

consistent with the results with no control variables, which further confirms the validity of the results. The results indicate that the ecological resilience of resource-based cities in the eastern region is weakly affected by government innovation preferences, while resource-based cities in the central and western regions will see a greater increase in ecological resilience under the effect of a reasonable increase in the level of government innovation preferences. However, excessive government innovation preferences can inhibit the development of ecological resilience, and the impact is higher in the western region than in the eastern region. The reason may be that resource-based cities in Eastern China are more oriented toward the development of high-tech industries and services than resource-based cities in the central and western regions of the country. Additionally, resource-based cities in Eastern China have basically eliminated the economic development model that relies on the exploitation of resources for the development of industries and manufacturing industries, leading to a higher level of ecological resilience of resource-based cities in the eastern region of the country. Therefore, the impact of government innovation preferences is smaller.

*(2) Heterogeneity of city size.* City size is a measure of the number of permanent residents in the urban area of a region. It not only reflects the ratio between the population in the urban area and the resident population of the region, i.e., the level of urbanization, but also reflects the volume of the economy, the attraction of employment and talent, and many other factors affecting the resilience of urban development. Thus, this paper refers to the classification of the size of cities in the report on the Seventh Population Census of 2020 in China by county (in which Type I and Type II large cities are defined as large cities, and Type I small cities and Type II small cities are defined as small cities). The paper conducts a regression analysis of the mechanism of the impact of government innovation preferences on the ecological resilience of resource-based cities, and the results are shown in Table 5: (1), (3), and (5) represent the regression results of large cities, medium-sized cities, and small cities without control variables, respectively; (2), (4) and (6) represent the regression results of large cities, medium-sized cities and small cities after adding control variables, respectively.

As shown in Table 6, in the regression results with no control variables, the primary term of government innovation preferences for large cities (0.02039***), medium-sized cities (0.01685***) and small cities (0.01549***) has a positive impact on the ecological resilience of resource-based cities. The impact coefficients are as follows (from high to low): large cities, medium-sized cities, and small cities. The coefficient of the secondary term of the impact of government innovation preferences on the ecological resilience of resource-based cities is negative, and the absolute value is as follows (from large to small): small cities (-0.00049***), medium-sized cities (-0.00046***) and large cities (-0.00038***). When the control variables are added, the impact of government innovation preferences on the ecological resilience of small resource-based cities is not significant, and the primary term of government innovation preferences is as follows: small cities (-0.00046***), medium-sized cities (-0.00048***) and large cities (-0.00038***). The primary term coefficient is higher for medium-sized cities (0.00766**) than for large cities (0.00613*), and the absolute value of the coefficient of the secondary term coefficient is higher for medium-sized cities (-0.00029**) than for large cities (-0.00016*). These results show that the ecological resilience of small resource-based cities is weakly affected by the role of government innovation preferences, and the ecological resilience of large resource-based cities and medium-sized resource-based cities will be enhanced with the strengthening of appropriate government innovation preferences. However, the marginal benefits will be reduced and show an unstable trend, and the instability of large resource-based cities will be higher than that of medium-sized resource-based cities. The reason may be that small resource-based cities usually do not have the basic conditions for realizing innovative practices and applying relevant innovations to environmental governance, resulting in the

**Table 6. Results of regression analysis of city size heterogeneity.**

| Variables | (1) | (2) | (3) | (4) | (5) | (6) |
|---|---|---|---|---|---|---|
| INP | 0.02039*** | 0.00613* | 0.01685*** | 0.00766** | 0.01549*** | 0.00193 |
|  | (5.64) | (2.04) | (4.94) | (2.12) | (7.29) | (0.78) |
| INP_2 | -0.00038*** | -0.00016* | -0.00046*** | -0.00029** | -0.00049*** | -0.00004 |
|  | (-3.32) | (-1.73) | (-3.42) | (-2.36) | (-6.33) | (-0.46) |
| PGDP |  | 0.01224*** |  | 0.00723** |  | 0.01141*** |
|  |  | (2.81) |  | (2.07) |  | (3.83) |
| GOV |  | -3.16219** |  | -3.11184 |  | 2.24233 |
|  |  | (-2.32) |  | (-1.15) |  | (1.44) |
| SIA |  | 0.01772 |  | 0.00421*** |  | -0.01553 |
|  |  | (0.70) |  | (3.72) |  | (-1.07) |
| FIL |  | 0.00000*** |  | 0.00000* |  | 0.00000 |
|  |  | (5.74) |  | (1.81) |  | (0.95) |
| Constant | 0.33027*** | 0.25906*** | 0.23768*** | 0.21291*** | 0.21164*** | 0.19551*** |
|  | (30.43) | (4.59) | (30.46) | (9.04) | (77.92) | (6.04) |
| Obs | 300 | 300 | 564 | 564 | 492 | 492 |
| Number of id | 25 | 25 | 47 | 47 | 41 | 41 |
| R − squared | 0.375 | 0.632 | 0.256 | 0.434 | 0.134 | 0.332 |
| Company FE | YES | YES | YES | YES | YES | YES |
| Control variables | NO | YES | NO | YES | NO | YES |
| F test | 0 | 0 | 0 | 0 | 0 | 0 |
| r2_a | 0.371 | 0.624 | 0.253 | 0.428 | 0.130 | 0.324 |
| F | 430.9 | 19.48 | 29.78 | 33.02 | 30.99 | 32.00 |

weak effect of government innovative preferences on ecological resilience. Large resource-based cities are better able to create such conditions to increase the level of ecological resilience, but at the same time, the complex organizational structure of these cities can also lead to the disadvantage of unstable growth in the level of ecological resilience.

**4.3.4. Robustness test.** (1) Replacement of the explanatory variables: To further verify the robustness of the benchmark regression model, this study first adopts the method of replacing the core explanatory variables to re-estimate the benchmark regression model. This paper introduces the number of patent applications (*PAP*) in each resource-based city to replace the proportion of financial expenditure on science and technology in GDP as a measure of government innovation preferences. The regression results are shown in Table 7 (1), in which the regression coefficients of the primary and secondary terms are 0.10856*** and -0.02738***, respectively. They are in the same direction as the original baseline regression results, and both of these coefficients are significant at the 1% level, verifying the robustness of the original benchmark regression results.

(2) Transformation of the regression model: Based on the entropy value method used in this study to calculate the index of the ecological resilience of resource-based cities, the results are all truncated data in the interval [0,1] so that they can be re-estimated by using the Tobit model. The results are shown in Table 7 (2). Both the primary term impact coefficient of government innovation preferences (0.00499***) and the quadratic term impact coefficient (-0.00013***) are consistent with the positive and negative directions of the benchmark regression results and are significant at the 1% level. These results demonstrate the robustness of the benchmark regression results of the study and further support the validity of the conclusion

**Table 7. Robustness test.**

| Variables | (1) | (2) | (3) |
|---|---|---|---|
| INP | | 0.00499*** | |
| | | (4.36) | |
| INP_2 | | -0.00013*** | |
| | | (-3.08) | |
| L.INP | | | 0.00769*** |
| | | | (3.44) |
| L.INP | | | -0.00024*** |
| | | | (-3.58) |
| PAP | 0.10856*** | | |
| | (4.62) | | |
| PAP_2 | -0.02738*** | -0.14205 | |
| | (-4.88) | (-0.22) | |
| Constant | 0.21037*** | 0.20387*** | 0.23169*** |
| | (18.28) | (21.70) | (13.34) |
| Obs | 1,243 | 1,243 | 1,243 |
| Number of id | 113 | 113 | 113 |
| R – squared | 0.478 | 0.353 | 0.422 |
| sigma_u | | 0.06696*** | |
| | | (14.42) | |
| sigma_e | | 0.03394*** | |
| | | (49.76) | |
| Company FE | YES | YES | YES |
| Control variables | YES | YES | YES |
| F test | 0 | 0 | 0 |
| r2_a | 0.476 | . | 0.419 |
| F | . | . | 37.69 |

that an increase in the level of governmental innovation preferences can promote the ecological resilience of resource-based cities; however, the marginal benefits will decline.

(3) Variable lag estimation: In this study, considering that the impact of government innovation preferences on the ecological resilience of resource-based cities may have a lag, the core explanatory variables are re-estimated with a lag. The results are shown in Table 7 (3). The coefficients of the primary and secondary terms of the impact of government innovation preferences on the ecological resilience of resource-based cities are (0.00769***) and (-0.00024***), respectively. Compared with the benchmark regression results of (0.01619***) and (-0.00036***), the direction remains consistent, but the absolute value of the results is large. The reason is perhaps that the government forms innovation preference consciousness, with financial science and technology expenditures related to the input of funds. However, the government also needs to form a certain innovation talent system and innovation practice system and then can carry out innovative activities to achieve innovative results, which will be applied to ecological governance to enhance ecological resilience and require a certain time span, leading to the existence of a lag. This result further proves the robustness of the benchmark regression.

**Table 8. Threshold effect test results under environmental decentralization.**

| Explanatory variable | Number of thresholds | F-value | P-value | threshold value | Threshold value | | |
|---|---|---|---|---|---|---|---|
| | | | | | Crit10 | Crit5 | Crit1 |
| Government innovation preferences | Single threshold | 41.86 | 0.0333 | 2.3993 | 29.4157 | 35.9394 | 64.5944 |
| | Double threshold | 30.51 | 0.1833 | 0.8721 | 40.4465 | 56.5398 | 78.2929 |
| | Triple threshold | 13.26 | 0.7467 | 2.1628 | 38.526 | 54.8298 | 82.5873 |

## 4.4. Further analysis: Threshold effect analysis

To further explore whether environmental decentralization can have a threshold effect on the mechanism of the impact of government innovation preferences on the ecological resilience of resource-based cities in this study, we first adopt the bootstrap method to test the likelihood ratio (LR) statistic, p value, critical value and corresponding interval 300 times to determine the number of thresholds and threshold values. The results are shown in Table 8. Simultaneously, we draw an LR diagram of the impact of the environmental decentralization threshold, as shown in Fig 4.

The results show that the single threshold value of environmental decentralization is significant at the 5% level. However, neither the double threshold nor the triple threshold is significant, which proves that there is a single threshold effect of environmental decentralization on the mechanism of the impact of government innovation preferences on the ecological resilience of resource-based cities. After determining the number of thresholds and threshold values, this study further conducts regression analysis using the threshold model, and the results are shown in Table 9.

From Table 9, it can be concluded that regarding the mechanism of the impact of government innovation preferences on the ecological resilience of resource-based cities, when the level of environmental decentralization is lower than the threshold value of 2.3993, the primary

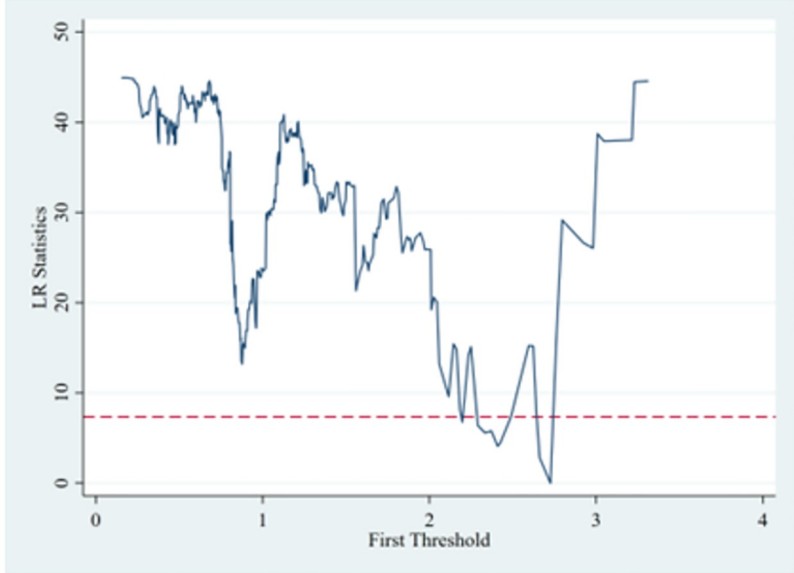

**Fig 4. Threshold effects of environmental decentralization LR map.**

**Table 9. Threshold effect regression results under environmental decentralization.**

| Variables | Estimated value | t-vaule | P-vaule |
|---|---|---|---|
| PGDP | 0.00997*** | 4.79 | 0.000 |
| GOV | -0.74214 | -0.74 | 0.461 |
| SIA | 0.00352** | 2.37 | 0.020 |
| FIL | 0.00000*** | 3.59 | 0.000 |
| $INP(END < 2.3993)$ | 0.00479** | 2.27 | 0.025 |
| $INP(END \geq 2.3993)$ | -0.03025 | -1.12 | 0.263 |
| $INP\_2(END < 2.3993)$ | -0.00012* | -1.84 | 0.068 |
| $INP\_2(END \geq 2.3993)$ | 0.00361 | 1.11 | 0.269 |
| _cons | 0.20704*** | 23.65 | 0.000 |
| N | 1356 | 1356 | 1356 |

term (0.00479**) and the secondary term (-0.00012*) of government innovation preferences have a positive effect and a negative effect on the level of ecological resilience of resource-based cities, respectively. Additionally both terms are significant at least at the 10% level. However, when the level of environmental decentralization crosses the threshold value of 2.3993, the effect of government innovation preferences on the ecological resilience of resource-based cities is not significant. The reason for this result is perhaps that when the level of environmental decentralization is low, the decision-making power regarding environmental governance-related issues is more in the hands of the central government, the government grasps the sustainability and reasonableness of ecological and environmental governance in resource-based cities more comprehensively, and the results related to government innovation preferences can better enhance the efficiency of ecological governance and further improve the level of ecological resilience. As the level of environmental decentralization increases, local governments have a stronger say in local environmental governance, and due to the pressure of short-term performance and the preference for local ecological governance, they may produce a lack of sustainability in ecological resilience and carry out local ecological governance at the expense of neighboring ecosystems. This will lead to the result that government innovation preferences cannot significantly influence the ecological resilience of resource-based cities.

## 5. Conclusion

This study selects relevant data covering 113 resource-based cities in China from 2009 to 2020, calculates the level of ecological resilience of such cities based on the entropy value method, and explores the direct and marginal effects of government innovation preferences on the ecological resilience of resource-based cities by using a two-way fixed effect model. Meanwhile, the mediating and threshold mechanisms of the impact of government innovation preferences on the ecological resilience of resource-based cities are analyzed with the educational level as the mediating variable and environmental decentralization as the threshold variable.

1. Analyzing the overall development history, the level of ecological resilience of China's resource-based cities has been increasing, fully affirming the series of improvement measures that China has developed to ensure that resource-based cities can escape the "resource curse". However, Daqing and Mudanjiang cities in Heilongjiang Province are still experiencing a downward trend in ecological resilience, and the marginal benefits of ecological resilience-related measures are declining. To realize the full coverage and sustainability of ecological resilience, relevant departments need to formulate targeted measures based on the development stage of resource-based cities and changes in the social situation.

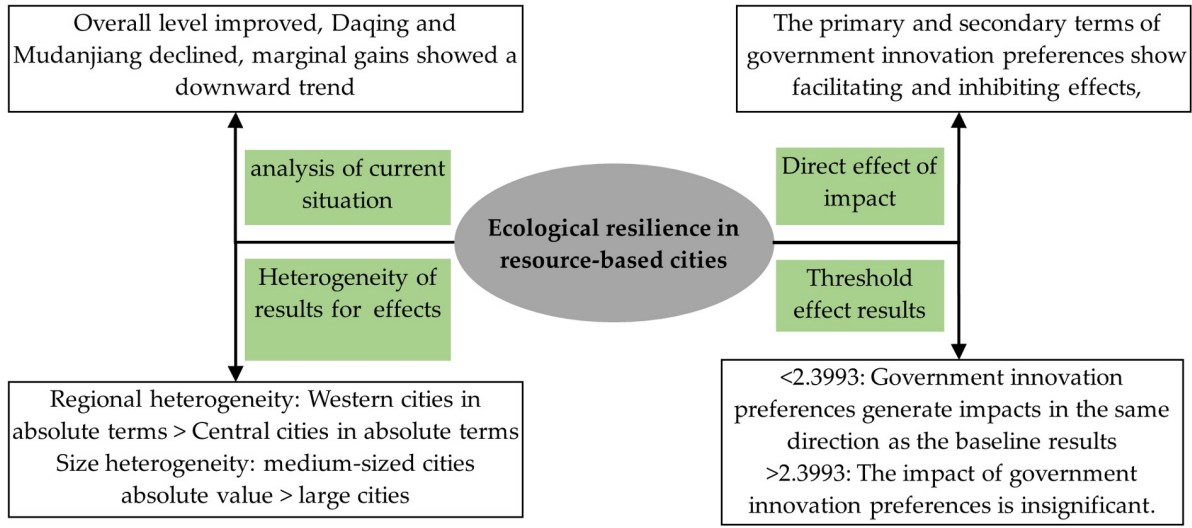

**Fig 5. Schematic of the conclusions.**

2. The primary term of government innovation preferences (0.00510**) has a significant positive effect on the ecological resilience of resource-based cities at the 5% level. The quadratic term of government innovation preferences (-0.00013**) has a significant negative effect on the ecological resilience of resource-based cities at the 5% level. These results suggest that reasonable government innovation preferences can enhance the ecological resilience of resource-based cities, but excessive innovation preferences may damage the ecological resilience of such cities. The educational level has a mediating effect on the mechanism of the impact of government innovation preferences on the ecological resilience of resource-based cities.

3. From the perspective of regional heterogeneity, the impact of the primary term coefficient of government innovation preferences on the ecological resilience of western resource-based cities (0.01564***) is higher than that of central resource-based cities (0.00682***), and the absolute value of the secondary term coefficient of the coefficient of the impact of western resource-based cities (-0.00053***) is higher than that of central resource-based cities (-0.00013*). From the perspective of city size heterogeneity, the primary term coefficient of government innovation preferences is higher for medium-sized resource-based cities (0.00766**) than for large resource-based cities (0.00613*), and the absolute value of the impact of the secondary term coefficient for medium-sized resource-based cities (-0.00029**) is greater than that for large resource-based cities (-0.00016*).

4. Under the threshold effect of environmental decentralization, when the level of environmental decentralization is lower than 2.3993, both the primary term (0.00479**) and the secondary term (-0.00012*) of government innovation preferences have significant positive and negative effects on the ecological resilience of resource-based cities at least at the 10% level, respectively. When the level of environmental decentralization crosses the threshold value of 2.3993, government innovation preferences have no significant effect on the ecological resilience of resource-based cities.

Specific conclusions are shown schematically in Fig 5.

## Supporting information

**S1 File. Data.**
(XLSX)

**S2 File. Code.**
(DO)

## Author Contributions

**Conceptualization:** Jie Zhang, Jiahui Yang, Feng Zhao.

**Data curation:** Jiahui Yang, Feng Zhao.

**Formal analysis:** Jiahui Yang, Feng Zhao.

**Funding acquisition:** Jie Zhang.

**Investigation:** Jie Zhang.

**Methodology:** Jie Zhang, Jiahui Yang, Feng Zhao.

**Project administration:** Jie Zhang, Jiahui Yang.

**Resources:** Jie Zhang.

**Software:** Jiahui Yang.

**Supervision:** Jie Zhang.

**Validation:** Jie Zhang, Jiahui Yang.

**Visualization:** Jie Zhang, Jiahui Yang.

**Writing – original draft:** Jie Zhang, Jiahui Yang, Feng Zhao.

**Writing – review & editing:** Jie Zhang, Feng Zhao.

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
