## [Decision Letter · Decision Letter 0]

24 Oct 2023

PONE-D-23-31556Do Government Innovation Preferences Enhance Ecological Resilience in Resource-based Cities under the Threshold of Environmental Decentralization?-- Empirical Evidence from 113 Resource-based Cities in ChinaPLOS ONE

Dear Dr. Yang,

Thank you for submitting your manuscript to PLOS ONE. After careful consideration, we feel that it has merit but does not fully meet PLOS ONE’s publication criteria as it currently stands. Therefore, we invite you to submit a revised version of the manuscript that addresses the points raised during the review process.

We look forward to receiving your revised manuscript.

Kind regards,

Fuyou Guo, (Ph.D.

Academic Editor

PLOS ONE

Journal Requirements:

   " This work was supported by the National Social Science Fund of China (No.22BTJ071)."

6. We note that Figures 3 and 5 in your submission contain map/satellite images which may be copyrighted. All PLOS content is published under the Creative Commons Attribution License (CC BY 4.0), which means that the manuscript, images, and Supporting Information files will be freely available online, and any third party is permitted to access, download, copy, distribute, and use these materials in any way, even commercially, with proper attribution. For these reasons, we cannot publish previously copyrighted maps or satellite images created using proprietary data, such as Google software (Google Maps, Street View, and Earth). For more information, see our copyright guidelines: http://journals.plos.org/plosone/s/licenses-and-copyright.

a. You may seek permission from the original copyright holder of Figure(s) [#] to publish the content specifically under the CC BY 4.0 license.  

7. Please remove your figures from within your manuscript file, leaving only the individual TIFF/EPS image files, uploaded separately. These will be automatically included in the reviewers’ PDF.

Additional Editor Comments:

Reviewer 1 Comments:

Government innovation preference is a solid foundation for carrying out innovation-driven strategies and an important guarantee for realizing efficient production modes. The structure of the article is clear and the process is explained well here. However, it could be improved in some detail.

1.Fig 1 is not necessary because you already give description in text.

2.The contributions of this paper require further clarification and improvement. It is suggested to further improve the innovation points.

3.Some of the statements are written without reference; try to add reference with every statement in your paper. Why you use the threshold effects model?

4. In the discussion part, the author should further explain the reasons for these results.

5. The overall quality of English is good, but need to be checked carefully again. I suggest the authors should look for an English native speaker to further check the language of the paper.

6. Some fresh paper can be used as ref. eg:

Tariq, G., Sun, H., Ali, I. et al. Influence of green technology, green energy consumption, energy efficiency, trade, economic development and FDI on climate change in South Asia. Sci Rep 12, 16376 (2022). https://doi.org/10.1038/s41598-022-20432-z.

Edziah B K., Sun H., Adom P K., Wang F., Agyemang A O., 2022. The role of exogenous technological factors and renewable energy in carbon dioxide emission reduction in Sub-Saharan Africa, Renewable Energy, 196: 1418–1428.

Reviewer 2 Comments:

This article studies the impact of government innovation preferences on the ecological resilience of resource-based cities, and explores the mediating effect of the level of education and the threshold effect of environmental decentralization. It has certain theoretical significance and practical implications. However, there are some serious problems in this article, and I have to make the Rejection decision.

1. The literature review of this article seems difficult to satisfy, as it did not attempt to comprehensively review the relevant literature in this field, and it's more like literature list. This is also the main reason why the article does not clearly elaborate on its literature contribution.

2. The authors do not clearly state the research motivation, research question, and research contribution of this article.

3. This article has a big problem in the standardization of format, such as the format of literature citation.

4. The authors do not refine the theoretical and practical implication of this article, but simply put forward the conclusion. These contents are obviously insufficient in the empirical articles.

5. The authors cite a large amount of Chinese journal literature, which is not recognized in an English journal submission.

6. The language of this article is relatively puerile, and need to find an English native-speaker for comprehensive language editing.

Reviewers' comments:

Reviewer's Responses to Questions

**Comments to the Author**

1. Is the manuscript technically sound, and do the data support the conclusions?

Reviewer #1: Partly

Reviewer #2: Yes

Reviewer #3: Yes

2. Has the statistical analysis been performed appropriately and rigorously? 

Reviewer #1: Yes

Reviewer #2: Yes

Reviewer #3: Yes

3. Have the authors made all data underlying the findings in their manuscript fully available?

Reviewer #1: Yes

Reviewer #2: Yes

Reviewer #3: Yes

4. Is the manuscript presented in an intelligible fashion and written in standard English?

Reviewer #1: No

Reviewer #2: Yes

Reviewer #3: Yes

5. Review Comments to the Author

Reviewer #1: This article studies the impact of government innovation preferences on the ecological resilience of resource-based cities, and explores the mediating effect of the level of education and the threshold effect of environmental decentralization. It has certain theoretical significance and practical implications. However, there are some serious problems in this article, and I have to make the Rejection decision.

1. The literature review of this article seems difficult to satisfy, as it did not attempt to comprehensively review the relevant literature in this field, and it's more like literature list. This is also the main reason why the article does not clearly elaborate on its literature contribution.

2. The authors do not clearly state the research motivation, research question, and research contribution of this article.

3. This article has a big problem in the standardization of format, such as the format of literature citation.

4. The authors do not refine the theoretical and practical implication of this article, but simply put forward the conclusion. These contents are obviously insufficient in the empirical articles.

5. The authors cite a large amount of Chinese journal literature, which is not recognized in an English journal submission.

6. The language of this article is relatively puerile, and need to find an English native-speaker for comprehensive language editing.

Reviewer #2: I have reviewed the revised version and found that the author(s) has(ve) addressed all the comments. However, I have no further suggestion. I hope that the comments are useful for this and also for upcoming projects.

Reviewer #3: Government innovation preference is a solid foundation for carrying out innovation-driven strategies and an important guarantee for realizing efficient production modes. The structure of the article is clear and the process is explained well here. However, it could be improved in some detail.

1.Fig 1 is not necessary because you already give description in text.

2.The contributions of this paper require further clarification and improvement. It is suggested to further improve the innovation points.

3.Some of the statements are written without reference; try to add reference with every statement in your paper. Why you use the threshold effects model?

4. In the discussion part, the author should further explain the reasons for these results.

5. The overall quality of English is good, but need to be checked carefully again. I suggest the authors should look for an English native speaker to further check the language of the paper.

6. Some fresh paper can be used as ref. eg:

Tariq, G., Sun, H., Ali, I. et al. Influence of green technology, green energy consumption, energy efficiency, trade, economic development and FDI on climate change in South Asia. Sci Rep 12, 16376 (2022). https://doi.org/10.1038/s41598-022-20432-z.

Edziah B K., Sun H., Adom P K., Wang F., Agyemang A O., 2022. The role of exogenous technological factors and renewable energy in carbon dioxide emission reduction in Sub-Saharan Africa, Renewable Energy, 196: 1418–1428.

6. PLOS authors have the option to publish the peer review history of their article (what does this mean?). If published, this will include your full peer review and any attached files.

Reviewer #1: No

Reviewer #2: No

Reviewer #3: No

---

## [Author Response · Author response to Decision Letter 0]

19 Dec 2023

Dear Editors,

Thank you and the reviewing experts for your review comments. Based on your and the reviewers' comments, I have revised the manuscript, responding to each point raised one by one, as described below:

Journal Requirements:

1. The new manuscript meets the style requirements of PLOS ONE.

2. The raw data of this manuscript were uploaded in the first submission.

3. The code for this manuscript was uploaded as an attachment in this submission.

4. The funder's role in the study is stated in the new cover letter.

5. The ORCID IDs of the corresponding authors are provided in the new submission.

6. Fig.3 and Fig.5 have been deleted from the new manuscript. new table and figure have been used for the replacements.

7. TIFF/EPS image files are uploaded in the new manuscript.

8. Material related to supporting information is added at the end of the new manuscript.

Reviewer 1's comments:

1. Figure 1 of the original manuscript has been removed from the new manuscript, and only the section that discusses the structure of the article has been retained.

2. The new manuscript adds the contribution of this study to future research in the Introduction section and further refines the innovations of the article.

3. The new manuscript refines the reference citation of the statement and further explains the reason for using the threshold effect model.

4. The new manuscript refines the explanation for the results presented in the Discussion section.

5. This manuscript has been revised and embellished for language by native English-speaking professionals.

6. Owing to the literature citation suggestions given by the reviewers, the new manuscript cites the references that the reviewers suggested using in the article.

Reviewer 2's comments.

1. In the new manuscript, the Literature Review section has been rewritten so that it covers as much as possible all areas of this study. The reasons for the contribution to the literature have also been refined.

2. The new manuscript clearly states the research motivation, research questions and research contributions.

3. The new manuscript has been revised based on the format of PLOS ONE journals, especially the citation format, which has been reorganized and rearranged.

4. The new manuscript adds the theoretical and practical significance of this study and explains the conclusions in more depth.

5. The new manuscript eliminates most of the Chinese references and selects English-language references as a supplement.

6. The new manuscript has been revised and embellished by native English-speaking professionals.

To facilitate your re-review of the manuscript, the revisions responding to the comments regarding the journal requirements are annotated in green, the revisions responding to the comments of reviewer 1 are annotated in red, the revisions responding to the comments of reviewer 2 are annotated in blue, and the revisions responding to the comments of the two reviewers are annotated in violet. Thank you again for the valuable comments, and we hope that you will inform us if you find any other deficiencies in the process of the re-reviewing the manuscript. Thank you again.

Yours sincerely,

Corresponding author: Jiahui Yang e-mail: 15136858260@163.com.

---

## [Decision Letter · Decision Letter 1]

4 Mar 2024

PONE-D-23-31556R1Do Government Innovation Preferences Enhance Ecological Resilience in Resource-based Cities under the Threshold of Environmental Decentralization?-- Empirical Evidence from 113 Resource-based Cities in ChinaPLOS ONE

Dear Dr. Yang,

Thank you for submitting your manuscript to PLOS ONE. After careful consideration, we feel that it has merit but does not fully meet PLOS ONE’s publication criteria as it currently stands. Therefore, we invite you to submit a revised version of the manuscript that addresses the points raised during the review process.

I have read the manuscript very carefully, it is an interesting work. Despite of this, I have comments that will be helpful to improve the quality of manuscript. The comments are as follows:

1- In abstract section, write the contribution point of the study.

2- The Abstract must report the aim of the study, the basic information on the sample (time span, countries analyzed), the empirical methodology used, the main findings, and the relevant policy implications.

3- Introduction and Literature Review should be split into two different sections.

4- The introduction section is missing the main contribution and objectives of the paper. A lot of similar paper is found by searching the literature. In this light, the author must have to provide the significance and main contributions of this study? Why an how this study is different from other studies?

5- section of literature review. The novelty of this paper should be further justified by highlighting main contributions to the existing literature. This could be clearly presented in the Literature review related work. Please consider please consider citing following papers entitled:

https://doi.org/10.1016/j.esr.2023.101216

https://doi.org/10.1111/1477-8947.12326

https://doi.org/10.1177/0958305X231181671

https://doi.org/10.1016/j.resourpol.2023.103553

https://doi.org/10.1007/S11356-022-24903-8/TABLES/7

https://doi.org/10.3390/su15010766

https://doi.org/https://doi.org/10.1016/j.strueco.2023.04.008

https://doi.org/10.1108/MEQ-11-2020-0275

6- The literature is missing.

7- Why the author used ecological footprint as an independent variable? Does this a correct choice taking with carbon emission as dependent variables?

8- I suggest to use the graphical representation of the results.

We look forward to receiving your revised manuscript.

Kind regards,

Fuyou Guo, (Ph.D.

Academic Editor

PLOS ONE

Additional Editor Comments (if provided):

I have read the manuscript very carefully, it is an interesting work. Despite of this, I have comments that will be helpful to improve the quality of manuscript. The comments are as follows:

1- In abstract section, write the contribution point of the study.

2- The Abstract must report the aim of the study, the basic information on the sample (time span, countries analyzed), the empirical methodology used, the main findings, and the relevant policy implications.

3- Introduction and Literature Review should be split into two different sections.

4- The introduction section is missing the main contribution and objectives of the paper. A lot of similar paper is found by searching the literature. In this light, the author must have to provide the significance and main contributions of this study? Why an how this study is different from other studies?

5- section of literature review. The novelty of this paper should be further justified by highlighting main contributions to the existing literature. This could be clearly presented in the Literature review related work. Please consider please consider citing following papers entitled:

https://doi.org/10.1016/j.esr.2023.101216

https://doi.org/10.1111/1477-8947.12326

https://doi.org/10.1177/0958305X231181671

https://doi.org/10.1016/j.resourpol.2023.103553

https://doi.org/10.1007/S11356-022-24903-8/TABLES/7

https://doi.org/10.3390/su15010766

https://doi.org/https://doi.org/10.1016/j.strueco.2023.04.008

https://doi.org/10.1108/MEQ-11-2020-0275

6- The literature is missing.

7- Why the author used ecological footprint as an independent variable? Does this a correct choice taking with carbon emission as dependent variables?

8- I suggest to use the graphical representation of the results.

Reviewers' comments:

Reviewer's Responses to Questions

**Comments to the Author**

1. If the authors have adequately addressed your comments raised in a previous round of review and you feel that this manuscript is now acceptable for publication, you may indicate that here to bypass the “Comments to the Author” section, enter your conflict of interest statement in the “Confidential to Editor” section, and submit your "Accept" recommendation.

Reviewer #2: (No Response)

Reviewer #3: All comments have been addressed

2. Is the manuscript technically sound, and do the data support the conclusions?

Reviewer #2: Yes

Reviewer #3: Yes

3. Has the statistical analysis been performed appropriately and rigorously? 

Reviewer #2: Yes

Reviewer #3: Yes

4. Have the authors made all data underlying the findings in their manuscript fully available?

Reviewer #2: Yes

Reviewer #3: Yes

5. Is the manuscript presented in an intelligible fashion and written in standard English?

Reviewer #2: Yes

Reviewer #3: Yes

6. Review Comments to the Author

Reviewer #2: I have read the manuscript very carefully, it is an interesting work. Despite of this, I have comments that will be helpful to improve the quality of manuscript. The comments are as follows:

1- In abstract section, write the contribution point of the study.

2- The Abstract must report the aim of the study, the basic information on the sample (time span, countries analyzed), the empirical methodology used, the main findings, and the relevant policy implications.

3- Introduction and Literature Review should be split into two different sections.

4- The introduction section is missing the main contribution and objectives of the paper. A lot of similar paper is found by searching the literature. In this light, the author must have to provide the significance and main contributions of this study? Why an how this study is different from other studies?

5- section of literature review. The novelty of this paper should be further justified by highlighting main contributions to the existing literature. This could be clearly presented in the Literature review related work. Please consider please consider citing following papers entitled:

https://doi.org/10.1016/j.esr.2023.101216

https://doi.org/10.1111/1477-8947.12326

https://doi.org/10.1177/0958305X231181671

https://doi.org/10.1016/j.resourpol.2023.103553

https://doi.org/10.1007/S11356-022-24903-8/TABLES/7

https://doi.org/10.3390/su15010766

https://doi.org/https://doi.org/10.1016/j.strueco.2023.04.008

https://doi.org/10.1108/MEQ-11-2020-0275

6- The literature is missing.

7- Why the author used ecological footprint as an independent variable? Does this a correct choice taking with carbon emission as dependent variables?

8- I suggest to use the graphical representation of the results.

Reviewer #3: Now all points have been addressed already after my careful checking. The structure of the paper is logical with good writing. so I suggest to accept it.

7. PLOS authors have the option to publish the peer review history of their article (what does this mean?). If published, this will include your full peer review and any attached files.

Reviewer #2: No

Reviewer #3: No

---

## [Author Response · Author response to Decision Letter 1]

13 Mar 2024

Dear Editor and reviewers:

Thank you very much for the insightful comments and suggestions. We have made corresponding revisions based on their advice. The words in blue are the changes that we made in the text.

The following are the responses and revisions that we made in response to the reviewers' questions and suggestions on an item-by-item basis.

Reviewer 2:

1. The new manuscript clarifies the contributions of this study in the abstract.

2. The new manuscript supplements and improves the abstract by including the research objectives, sample information, empirical methodology, main conclusions, and policy implications of this study.

3. As the overall structure of the new manuscript, Part I is the Introduction; Part II is the Literature Review; Part III is the Research hypotheses and model construction; Part IV is the Empirical analysis; and Part V is the Conclusion.

4. The new manuscript has added the significance and main contributions of this study to the Introduction section. The innovations in the Literature Review section explain the difference between this study and existing studies.

5. The new manuscript further enriches the Literature Review section of this study by referring to some of the references recommended by the reviewers.

6 The new manuscript has added to and improved the Literature Review section.

7 The main purpose of this study is to investigate the relationship between government innovation preferences and ecological resilience in resource-based cities from the perspective of environmental decentralization. It does not include the relevant variables of the ecological footprint and carbon emissions.

8. Figure 5 has been added to the new manuscript to represent the results based on the findings.

Reviewer 3:

We are very grateful to Reviewer 3 for recognizing our paper.

Once again, thank you for your valuable comments. We hope that you will let us know if you and the reviewers find any other deficiencies during the review process.

Yours sincerely,

Corresponding author: Jiahui Yang, E-mail: 15136858260@163.com

---

## [Decision Letter · Decision Letter 2]

9 Apr 2024

PONE-D-23-31556R2Do Government Innovation Preferences Enhance Ecological Resilience in Resource-based Cities under the Threshold of Environmental Decentralization?-- Empirical Evidence from 113 Resource-based Cities in ChinaPLOS ONE

Dear Dr. Yang,

Thank you for submitting your manuscript to PLOS ONE. After careful consideration, we feel that it has merit but does not fully meet PLOS ONE’s publication criteria as it currently stands. Therefore, we invite you to submit a revised version of the manuscript that addresses the points raised during the review process.

This paper establishes a mediating effect model and a threshold effect model and empirically analyzes 2009-2020 panel data covering 113 resourcebased cities in China as an example. Now the revisions are all ok.

1. For the current topic, it can be further optimized and simplified.

2. The format of the entire text needs to be unified according to the needs of the publication, for example, some literature does not have a unified citation format, and for example, reference 47 needs to be adjusted.

We look forward to receiving your revised manuscript.

Kind regards,

Fuyou Guo, (Ph.D.

Academic Editor

PLOS ONE

Journal Requirements:

Additional Editor Comments:

This paper establishes a mediating effect model and a threshold effect model and empirically analyzes 2009-2020 panel data covering 113 resourcebased cities in China as an example. Now the revisions are all ok.

1. For the current topic, it can be further optimized and simplified.

2. The format of the entire text needs to be unified according to the needs of the publication, for example, some literature does not have a unified citation format, and for example, reference 47 needs to be adjusted.

Reviewers' comments:

Reviewer's Responses to Questions

**Comments to the Author**

1. If the authors have adequately addressed your comments raised in a previous round of review and you feel that this manuscript is now acceptable for publication, you may indicate that here to bypass the “Comments to the Author” section, enter your conflict of interest statement in the “Confidential to Editor” section, and submit your "Accept" recommendation.

Reviewer #3: All comments have been addressed

2. Is the manuscript technically sound, and do the data support the conclusions?

Reviewer #3: Yes

3. Has the statistical analysis been performed appropriately and rigorously? 

Reviewer #3: Yes

4. Have the authors made all data underlying the findings in their manuscript fully available?

Reviewer #3: Yes

5. Is the manuscript presented in an intelligible fashion and written in standard English?

Reviewer #3: Yes

6. Review Comments to the Author

Reviewer #3: This paper establishes a mediating effect model and a threshold effect model and empirically analyzes 2009-2020 panel data covering 113 resourcebased cities in China as an example. Now the revisions are all ok.

1. For the current topic, it can be further optimized and simplified.

2. The format of the entire text needs to be unified according to the needs of the publication, for example, some literature does not have a unified citation format, and for example, reference 47 needs to be adjusted.

7. PLOS authors have the option to publish the peer review history of their article (what does this mean?). If published, this will include your full peer review and any attached files.

Reviewer #3: No

---

## [Author Response · Author response to Decision Letter 2]

23 Apr 2024

Dear Editor and reviewers:

Thank you very much for the insightful comments and suggestions. We have made corresponding revisions based on their advice. The words in blue are the changes that we made in the text.

The following are the responses and revisions that we made in response to the reviewers' questions and suggestions on an item-by-item basis.

Reviewer 3:

1. The new manuscript optimizes and simplifies the topic of this paper by revising it to “Do Government Innovation Preferences Enhance Ecological Resilience in Resource-Based Cities?--Based on mediating effect and threshold effect perspectives”.

2. The new manuscript carefully proofreads references and standardizes the format of reference notation.

Once again, thank you for your valuable comments. We hope that you will let us know if you and the reviewers find any other deficiencies during the review process.

Yours sincerely,

Corresponding author: Jiahui Yang, E-mail: 15136858260@163.com

---

## [Decision Letter · Decision Letter 3]

30 Apr 2024

Do Government Innovation Preferences Enhance Ecological Resilience in Resource-Based Cities?--Based on mediating effect and threshold effect perspectives

PONE-D-23-31556R3

Dear Dr. Yang,

We’re pleased to inform you that your manuscript has been judged scientifically suitable for publication and will be formally accepted for publication once it meets all outstanding technical requirements.

Kind regards,

Xingwei Li, Ph.D.

Academic Editor

PLOS ONE

Additional Editor Comments (optional):

Reviewers' comments:

Reviewer's Responses to Questions

**Comments to the Author**

1. If the authors have adequately addressed your comments raised in a previous round of review and you feel that this manuscript is now acceptable for publication, you may indicate that here to bypass the “Comments to the Author” section, enter your conflict of interest statement in the “Confidential to Editor” section, and submit your "Accept" recommendation.

Reviewer #3: All comments have been addressed

2. Is the manuscript technically sound, and do the data support the conclusions?

Reviewer #3: Yes

3. Has the statistical analysis been performed appropriately and rigorously? 

Reviewer #3: Yes

4. Have the authors made all data underlying the findings in their manuscript fully available?

Reviewer #3: Yes

5. Is the manuscript presented in an intelligible fashion and written in standard English?

Reviewer #3: Yes

6. Review Comments to the Author

Reviewer #3: This paper establishes a mediating effect model and a threshold effect model and empirically analyzes 2009-2020 panel data covering 113 resourcebased cities in China as an example. Now the revisions are all ok, so I suggest to acdept it.

7. PLOS authors have the option to publish the peer review history of their article (what does this mean?). If published, this will include your full peer review and any attached files.

Reviewer #3: No

---

## [Editor Report · Acceptance letter]

27 Jun 2024

PONE-D-23-31556R3 

PLOS ONE

Dear Dr. Yang, 

I'm pleased to inform you that your manuscript has been deemed suitable for publication in PLOS ONE. Congratulations! Your manuscript is now being handed over to our production team.

Kind regards, 

on behalf of

Prof. Dr. Xingwei Li 

Academic Editor

PLOS ONE